# Comparison of a Peripheral Nerve Block versus Spinal Anesthesia in Foot or Ankle Surgery: A Systematic Review and Meta-Analysis with a Trial Sequential Analysis

**DOI:** 10.3390/jpm13071096

**Published:** 2023-07-04

**Authors:** Myeongjong Lee, Cheol Lee, Junsung Lim, Hyungtae Kim, Yoo-Shin Choi, Hyun Kang

**Affiliations:** 1Department of Anesthesiology and Pain Medicine, Research Institute of Medical Science, Konkuk University School of Medicine, 82 Gugwondae-ro, Chungju 27376, Republic of Korea; gooddr21@kku.ac.kr; 2Department of Anesthesiology and Pain Medicine, Wonkwang University School of Medicine, 895 Muwang-ro, Iksan 54538, Republic of Korea; ironyii@wku.ac.kr (C.L.); sy4957@wkuh.org (J.L.); 3Department of Anesthesiology and Pain Medicine, Asan Medical Center, University of Ulsan College of Medicine, 88 Olympic-ro 43-gil, Songpa-gu, Seoul 05505, Republic of Korea; ingwei2475@gmail.com; 4Department of Surgery, Chung-Ang University College of Medicine, 84 Heukseok-ro, Dongjak-gu, Seoul 06911, Republic of Korea; choiys@cau.ac.kr; 5Department of Anesthesiology and Pain Medicine, Chung-Ang University College of Medicine, 84 Heukseok-ro, Dongjak-gu, Seoul 06911, Republic of Korea

**Keywords:** anesthesia, spinal, nerve block, orthopedic procedures, systematic review

## Abstract

**Background:** This systematic review and meta-analysis with trial sequential analysis (TSA) aimed to compare perioperative outcomes of peripheral nerve blocks (PNBs) and spinal anesthesia (SA) in elective foot and ankle surgery. **Methods:** The study protocol was registered in PROSPERO (CRD42021229597). Researchers independently searched PubMed, EMBASE, and the Cochrane Central Register of Controlled Trials for relevant randomized controlled trials (RCTs). **Results:** Analysis of nine RCTs (*n* = 802; 399 PNBs, 403 SA) revealed significantly shorter block performance times (WMD: 7.470; 95% CI 6.072 to 8.868), the onset of sensory (WMD: 7.483; 95% CI 2.837 to 12.130) and motor blocks (WMD: 9.071; 95% CI 4.049 to 14.094), durations of sensory (WMD: 458.53; 95% CI 328.296 to 588.765) and motor blocks (WMD: 247.416; 95% CI 95.625 to 399.208), and significantly higher postoperative analgesic requirements (SMD: −1.091; 95% CI −1.634 to −0.549) in the SA group. Additionally, systolic blood pressure (SBP) at 30 min (WMD: 13.950; 95% CI 4.603 to 23.298) was lower in the SA group. **Conclusions:** The SA demonstrated shorter block performance time, faster onset and shorter duration of sensory and motor blocks, higher postoperative analgesic requirements, and lower SBP at 30 min compared to PNBs in elective foot and ankle surgery.

## 1. Introduction

The use of peripheral nerve blocks (PNBs) in orthopedic limb surgery has become increasingly popular [1,2]. Commonly used peripheral nerve blocks for foot and/or ankle surgery are the sciatic, femoral, popliteal, lateral femoral cutaneous, saphenous, and ankle blocks [3,4]. As single-site injections, these techniques do not provide broad and adequate anesthetic coverage for foot and/or ankle surgery. Therefore, these techniques are traditionally not used as a single anesthetic technique for foot and/or ankle surgery but are combined with general anesthesia or spinal anesthesia for postoperative pain relief [5,6]. However, some studies have recently demonstrated that PNBs alone or combined can provide adequate anesthesia in foot and ankle surgery [7,8,9,10,11,12,13,14,15].

Previous studies have suggested that PNBs have considerable clinical advantages, such as less cardiovascular effects, a longer analgesic duration, lower hospital costs, a shorter length of stay (LOS), and a lower incidence of urinary retention and post-dural puncture headache (PDPH) [1,2]. However, technical difficulties, the potentials for inadequate blocks, increased block performance times, delayed block onsets, and additional needs for equipment such as a peripheral nerve stimulator or a portable ultrasound unit are regarded as obstacles to the application of PNBs in everyday clinical practice. Therefore, due to its easiness, short block performance time, onset time, and reliability for an adequate block, spinal anesthesia (SA) is more often performed in clinical practice.

There has been an attempt to incorporate the evidence from various studies investigating the effects of combinations of PNBs [16]. However, it incorporated studies with broad designs, such as retrospective series and prospective and retrospective cohort studies. Furthermore, as this analysis included effects from adjuvants, it did not focus on the pure effects of PNBs. To date, no systematic review and meta-analysis have compared the pure effects of PNBs with SA for foot and ankle surgery.

Therefore, we critically reviewed and synthesized the current evidence from randomized controlled trials (RCTs) to compare the effects and safety of SA versus PNBs in patients undergoing foot and ankle surgery.

## 2. Materials and Methods

We developed the protocol for this systematic review and meta-analysis with a trial sequential analysis according to the Preferred Reporting Items for Systematic Review and Meta-Analysis Protocol (PRISMA-P) and registered it in the PROSPERO network (registration number: CRD42021229597, available at https://www.crd.york.ac.uk/PROSPERO/display_record.php?RecordID=229597 (accessed on 14 June 2021).

This study was completed by observing the recommendations of the Cochrane Collaboration [17] and reported by following the PRISMA statement guidelines [18].

### 2.1. Inclusion and Exclusion Criteria

The inclusion and exclusion criteria were determined before conducting any systematic searches. We included full reports of RCTs investigating the efficacy, patient satisfaction, and adverse effects in patients undergoing foot or ankle surgery between PNBs and SA.

The PICO-SD information is as follows:

Patients (P): all elective patients undergoing foot or ankle surgery with PNBs and SA.

Intervention (I): PNBs (femoral, sciatic, popliteal) performed as anesthesia using a single dose or continuous infusion.

Comparison (C): SA.

Outcome measurements (O): The primary outcome of this systematic review and meta-analysis with a trial sequential analysis was the pain scores during surgery and the postoperative period. The secondary outcomes were patient satisfaction and adverse effects.

Study design (SD): The inclusion criteria for this systematic review and meta-analysis with a trial sequential analysis were the full reports of randomized controlled trials (RCTs). The exclusion criteria were observational studies, conference abstracts, posters, case reports, case series, comments or letters to the editor, reviews, and laboratory or animal studies.

### 2.2. Information Source and Search Strategy

To identify RCTs for this systematic review and meta-analysis with a trial sequential analysis, two investigators (ML and CL) independently performed searches of the PubMed, EMBASE, and Cochrane Central Register of Controlled Trials (CENTRAL) databases on 28 June 2021 and updated them on 15 March 2023. The search terms included the following in various combinations with free text, Medical Subject Headings, and EMTREE terms: (peripheral nerve block OR sciatic nerve block OR femoral nerve block OR popliteal nerve block OR saphenous nerve block OR ankle block) AND (spinal anesthesia OR spinal nerve block) AND (ankle surgery OR foot surgery). In addition, we searched the reference lists of original articles to ensure that we included all available studies. No limitations were placed on the publication date or language.

### 2.3. Study Selection

Two investigators (ML and CL) independently scanned the titles and abstracts of the reports identified. If a report was considered eligible from the title or abstract, the full text was retrieved and evaluated. All abstracts that could not provide sufficient information regarding the eligibility criteria were selected for full-text evaluation. Potentially relevant studies that at least one investigator identified were retrieved, and the full-text versions were evaluated. Articles that met the inclusion criteria were assessed separately by the two investigators (ML and CL), and any discrepancies were resolved through discussion. Disagreement over inclusion or exclusion was settled by a discussion with a third investigator (HK).

Kappa statistics were used to measure the degree of agreement for study selection between the two independent investigators. Kappa statistics were interpreted as follows: (1) less than 0, less than chance agreement; (2) 0.01 to 0.20, slight agreement; (3) 0.21 to 0.40, fair agreement; (4) 0.41 to 0.60, moderate agreement; (5) 0.61 to 0.80, substantial agreement; and (6) 0.8 to 0.99, almost perfect agreement [19].

### 2.4. Data Extraction

Using a standardized data collection form, two independent investigators (ML and CL) extracted all relevant data from the included studies, input them into standardized forms, and then crosschecked them. Any discrepancy was resolved through discussion. If an agreement could not be reached, a third investigator (HK) provided a resolution.

The extracted data included the first author, journal, publication year, country of origin, study protocol registration (registry and registration number), study design, characteristics and conversion to general anesthesia, intraoperative vital signs (systolic blood pressure (SBP), diastolic blood pressure (DBP), mean blood pressure, heart rate), block performance time, onset time and duration of sensory and motor blocks, postoperative analgesic requirements, all intervention-related side effects (urinary retention, PDPH, local anesthetic systemic toxicity (LAST), nerve damage), and patient satisfaction.

The data were initially extracted from tables or text. In cases involving missing or incomplete data, we tried to contact the authors to obtain the relevant information.

### 2.5. Risk of Bias

Risk of bias was assessed using the Revised Cochrane risk of bias tool for randomized trials (RoB 2.0) version (22 August 2019) by two independent authors (ML and HK). RoB 2.0 is structured into five domains: D1, bias arising from the randomization process; D2, bias because of deviations from the intended interventions; D3, bias because of missing outcome data; D4, discrimination in the measurement of the outcome; and D5, bias in the selection of the reported result. We also evaluated the overall risk of bias. The risk was judged as low risk when the risk of bias for all domains was low, high when the risk of bias for at least one domain was high, or the risk of biases for multiple domains was of some concern, and some concern if the overall judgment neither low nor high.

### 2.6. Data Analysis

#### 2.6.1. Conventional Meta-Analysis

A meta-analysis was conducted using Comprehensive Meta-Analysis version 2.0 (Englewood, NJ, USA, 2008). Two investigators (ML and HK) independently input all data into the software. The pooled risk ratio (RR) for binary variables or weighted mean difference (WMD) for quantitative data and their 95% confidence intervals (CIs) were calculated for each outcome. A random-effects model was used to account for clinical or methodological heterogeneity in each study. Statistical heterogeneity was assessed using the I^2^ test, with I^2^ > 50 indicating significant heterogeneity. We performed a sensitivity analysis to explore heterogeneity by removing one study at a time and evaluating whether it altered our results. Publication bias was not estimated since fewer than 10 studies were included. We calculated the number needed to treat (NNT) using a 95% CI based on the absolute risk reduction to estimate the overall clinical impact of the intervention.

#### 2.6.2. Trial Sequential Analysis

A conventional meta-analysis runs the risk of random errors due to sparse data. A trial sequential analysis (TSA) is a methodology that includes a required information size (RIS) calculation for a meta-analysis with a threshold for statistical significance, which controls the risk of potential false-positive and false-negative findings of meta-analyses [20]. Therefore, we additionally performed a TSA on the outcomes to calculate the RIS and assess whether our results were conclusive. We used a random-effects model with the DerSimonian–Laird (DL) to construct the cumulative Z curve. The TSA was performed to maintain an overall 5% risk of a type I error.

When the cumulative Z curve crossed the trial sequential monitoring boundary or entered the futility area, the sufficient level of evidence for accepting or rejecting the anticipated intervention effect may have been reached, and no further studies were needed. If the Z curve did not cross any boundaries and the RIS was not reached, the evidence to reach a conclusion was insufficient, indicating the requirement for more studies.

For dichotomous outcomes, we estimated the RIS based on the observed proportion of patients with an outcome in the PNB group (the cumulative proportion of patients with an event relative to all patients in the PNB group), a relative risk reduction of 30% in the SA group, an alpha of 5% for all our outcomes, a beta of 20%, and the observed diversity as suggested by the trials in the meta-analysis.

For quantitative outcomes, we used the observed standard deviation (SD) in the trial sequential analysis, a mean difference of the observed SD/3, an alpha of 5% for all outcomes, a beta of 20%, and the observed diversity as suggested by the trials in the meta-analysis.

#### 2.6.3. Quality of the Evidence

The evidence grade was determined using the guidelines of the Grading of Recommendations Assessment, Development, and Evaluation (GRADE) system, which uses a sequential assessment of the evidence quality, followed by an assessment of risk–benefit balance and a subsequent judgment on the strength of the recommendations [21].

## 3. Results

### 3.1. Study Selection

From the PubMed, EMBASE, and CENTRAL database searches, 162 studies were initially selected. After adjusting for duplicates (*n* = 44), 118 studies remained. Of these, 102 studies were excluded after reviewing the titles and abstracts, as they were not relevant. At this stage of study selection, the kappa value for selecting studies between the two reviewers was 0.759. Full texts of the remaining 16 studies were reviewed in detail. Of these, seven studies were further excluded because one study did not compare PNBs versus SA [22], and six were not RCTs [5,23,24,25,26,27]. The kappa value for selecting articles between the two investigators was 0.875.

Finally, nine studies satisfied the inclusion criteria and were incorporated into this study [7,8,9,10,11,12,13,14,15]. Two studies included below-knee amputation (BKA), and the authors of two studies provided additional data excluding BKA at our e-mail on request [10,15]. Thus, nine studies with a total of 802 patients were included in our meta-analysis (Figure 1).

### 3.2. Study Characteristics

The study characteristics are shown in Table 1. Seven studies performed ultrasound-guided PNBs with or without a nerve stimulator [7,8,9,10,11,13,14]. Two studies used a nerve stimulator alone [12,15]. Seven studies performed two PNBs (femoral or saphenous and sciatic nerve blocks) [7,9,10,11,12,13,15]. Dabir et al. performed three PNBs (femoral, lateral femoral cutaneous, and popliteal blocks), and Yang et al. performed four PNBs (femoral, obturator, lateral femoral cutaneous, and sacral plexus blocks). Only three trials were registered at ClinicalTrials.gov, Clinicaltrialsregister.eu, and the Iran Registry of Clinical Trials [7,8,10].

### 3.3. Risk of Bias

The risk of bias assessment performed using the Cochrane tool for the included studies is presented in Table 2. Studies were judged to have low risk [7,14], some concerns [8,9,11,12,13], or high risk [10,15]. Among the nine included studies, bias in the measurement of the outcome and bias in the selection of the reported results were assessed as “low risk”, except for Chauhan et al.’s study [15]; bias arising from the randomization process; bias due to deviations from intended intervention; and bias due to missing outcome data were assessed as “some concerns” in four studies, three studies, and two studies, respectively.

### 3.4. Conversion to General Anesthesia

A total of eight studies [7,8,9,10,11,13,14,15] (602 patients) measured the conversion rate to general anesthesia. There was no evidence of a difference in the conversion rate to general anesthesia between the PNB and SA groups (RR: 2.261; 95% CI 0.514 to 9.953; I^2^ = 0.0), but the conversion rate to general anesthesia was lower in the SA group (0.66%, 2 of 303) than in the PNB group (3.01%, 9 of 299) in terms of the NNT (NNT harm (NNTH): 43; 95% CI NNTH 22 to NNTH 478) (Figure 2, Table 3).

The TSA indicated that only 2.9% (602 of 20,771 patients) of the RIS was accrued. The trial sequential monitoring boundary was ignored due to too little information use. The cumulative Z curve did not cross the conventional test boundary (Appendix A, Table 3).

### 3.5. Block Performance Time

Four studies [8,9,13,14] (260 patients) measured the block performance time. The block performance time was significantly lower in the SA group than in the PNB group (WMD: 7.470; 95% CI 6.072 to 8.868; I^2^ = 88.66) (Figure 3, Table 3).

### 3.6. Onset Time of the Sensory Block

Four studies [8,13,14,15] (234 patients) measured the onset time of the sensory block. The onset time of the sensory block was significantly shorter in the SA group than in the PNB group (WMD: 7.483; 95% CI 2.837 to 12.130; I^2^ = 99.665 (Appendix A, Table 3).

A sensitivity analysis performed by removing the studies by Yang et al. [14] changed the statistical significance of results without eliminating heterogeneity (Appendix A).

The TSA indicated that only 18.9% (234 of 1236 patients) of the RIS was accrued. The cumulative Z curve crossed both the conventional test boundary and the trial sequential monitoring boundary (Appendix A, Table 3).

### 3.7. Onset Time of the Motor Block

Seven studies [8,9,11,12,13,14,15] (554 patients) measured the onset time of the motor block. The onset time of the motor block was significantly shorter in the SA group than in the PNB group (WMD: 9.071; 95% CI 4.049 to 14.094; I^2^ = 99.403) (Figure 4, Table 3).

A sensitivity analysis performed by removing one study at a time showed no change in statistical significance (Appendix A).

The TSA indicated that 93.7% (554 of 591 patients) of the RIS was accrued. The cumulative Z curve crossed both the conventional test boundary and the trial sequential monitoring boundary (Appendix A, Table 3).

### 3.8. Duration of the Sensory Block

Seven studies [7,10,11,12,13,14,15] (679 patients) measured the duration of the sensory block. The duration of the sensory block was significantly longer in the PNB group than in the SA group (WMD: 458.530; 95% CI 328.296 to 588.765; I^2^ = 97.02) (Figure 5, Table 3).

A sensitivity analysis performed by removing one study at a time showed no change in statistical significance (Appendix A).

The TSA indicated that the number of accrued patients exceeded the RIS (679 of 246 patients). The cumulative Z curve crossed both the conventional test boundary and the trial sequential monitoring boundary (Appendix A, Table 3).

### 3.9. Duration of the Motor Block

Four studies [12,13,14,15] (397 patients) measured the duration of the motor block. The duration of the motor block was significantly longer in the PNB group than in the SA group (WMD: 247.416; 95% CI 95.625 to 399.208; I^2^ = 97.325) (Appendix A, Table 3).

A sensitivity analysis performed by removing one study at a time showed no change in statistical significance (Appendix A).

The TSA indicated that 94.3% (397 of 421 patients) of the RIS was accrued. The cumulative Z curve crossed both the conventional test boundary and the trial sequential monitoring boundary (Figure 6, Table 3).

### 3.10. Postoperative Analgesic Requirements

Three studies [7,12,15] (407 patients) measured the postoperative analgesic requirements. The postoperative analgesic requirement was significantly lower in the PNB group than in the SA group (SMD: −1.091; 95% CI −1.634 to −0.549; I^2^ = 83.10) (Appendix A, Table 3).

A sensitivity analysis performed by removing one study at a time showed no change in statistical significance (Appendix A).

The TSA indicated that only 23.4% (407 of 1741 patients) of the RIS was accrued. The cumulative Z curve crossed the conventional test boundary and the trial sequential monitoring boundary (Appendix A, Table 3).

### 3.11. Incidence of Hypotension

A total of four studies [9,10,11,14] (272 patients) measured the incidence of hypotension. The incidence of hypotension was significantly lower in the PNB group than in the SA group. (RR: 0.152; 95% CI 0.042 to 0.548; I^2^ = 0.0; NNT benefit (NNTB): 7; 95% CI NNTB 5 to NNTH 14) (Appendix A, Table 3).

The TSA indicated that only 6.7% (272 of 4043 patients) of the RIS was accrued. The cumulative Z curve crossed the conventional test boundary but did not cross the trial sequential monitoring boundary (Appendix A, Table 3).

### 3.12. Use of Vasoactive Drugs

A total of three studies [7,9,10] (305 patients) measured the use of vasoactive drugs. The use of vasoactive drugs was significantly lower in the PNB group than in the SA group. (RR: 0.253; 95% CI 0.101 to 0.638; I^2^ = 0.0; NNTB: 8; 95% CI NNTB 5 to NNTH 15) (Appendix A, Table 3).

The TSA indicated that only 8.2% (305 of 3698 patients) of the RIS was accrued. The cumulative Z curve crossed the conventional test boundary but did not cross the trial sequential monitoring boundary (Appendix A, Table 3).

### 3.13. Systolic Blood Pressure

Three studies [10,13,15] (212 patients) measured SBP. There was no evidence of a difference between the SA and PNB groups at T0. (WMD: −0.112; 95% CI −5.329 to 5.168; I^2^ = 31.501) (Appendix A, Table 3).

The SBP was significantly lower in the SA group than in the PNB group at T30. (WMD: 13.950; 95%CI 4.603 to 23.298; I^2^ = 74.829) (Appendix A, Table 3).

A sensitivity analysis performed by removing the studies by Xu et al. [13] changed the statistical significance of results without eliminating heterogeneity (Appendix A).

The TSA indicated that only 11.1% (212 of 1911 patients) of the RIS was accrued at T0. The cumulative Z curve crossed the conventional test boundary but did not cross the trial sequential monitoring boundary (Appendix A, Table 3). The TSA indicated that the number of accrued patients exceeded the RIS (212 of 75 patients) at T30. The cumulative Z curve crossed both the conventional test boundary and the trial sequential monitoring boundary (Appendix A, Table 3).

### 3.14. Diastolic Blood Pressure

Three studies [10,13,15] (212 patients) measured the DBP. There was no evidence of a difference between the SA and PNB groups at T0. (WMD: −1.297; 95% CI −3.931 to 1.337; I^2^ = 0.0) (Appendix A, Table 3).

There was no evidence of a difference between the SA and PNB groups at T30. (WMD: 4.535; 95% CI −0.969 to 10.039; I^2^ = 75.674) (Appendix A, Table 3).

The TSA indicated that only 11.8% (212 of 1802 patients) of the RIS was accrued at T0. The cumulative Z curve did not cross the conventional test boundary. (Appendix A, Table 3). The TSA indicated that only 32.8% (212 of 646 patients) of the RIS was accrued at T30. The cumulative Z curve did not cross the conventional test boundary (Appendix A, Table 3).

### 3.15. Heart Rate

Three studies [10,13,15] (212 patients) measured the heart rate. There was no evidence of a difference between the SA and PNB groups at T0. (WMD: −0.204; 95% CI −3.696 to 3.288; I^2^ = 0.062) (Appendix A, Table 3).

There was no evidence of a difference between the SA and PNB groups at T30. (WMD: 2.617; 95% CI −3.265 to 8.599; I^2^ = 66.401) (Appendix A, Table 3).

The TSA indicated that only 11.8% (212 of 1802 patients) of the RIS was accrued at T0. The cumulative Z curve did not cross the conventional test boundary (Appendix A, Table 3). The TSA indicated that only 9.5% (212 of 2237 patients) of the RIS was accrued at T30. The cumulative Z curve did not cross the conventional test boundary (Appendix A, Table 3).

### 3.16. Urinary Retention

A total of four studies [9,11,12,15] (354 patients) measured urinary retention. There was no evidence of a difference between the SA and PNB groups (RR: 0.089; 95% CI 0.002 to 3.297; I^2^ = 0.0), but urinary retention occurred less frequently in the PNB group (0.0%, 0 of 175) than in the SA group (5.59%, 10 of 179) in terms of the NNT (NNTB: 18; 95% CI NNTB 11 to NNTH 45) (Appendix A, Table 3).

The TSA indicated that only 18.5% (354 of 1912 patients) of the RIS was accrued. The cumulative Z curve did not cross the conventional test boundary (Appendix A, Table 3).

### 3.17. Post-Dural Puncture Headache

A total of four studies [9,11,14,15] (234 patients) measured the incidence of PDPH. There was no evidence of a difference between the SA and PNB groups. (RR: 0.159; 95% CI 0.004 to 6.071; I^2^ = 0.0), but the incidence of PDPH was lower in the PNB group (0.00%, 0 of 115) than in the SA group (3.36%, 4 of 119) in terms of the NNT (NNTB: 30; 95% CI NNTB 15 to NNTH 813) (Appendix A, Table 3).

The TSA indicated that only 1.2% (234 of 14,721 patients) of the RIS was accrued. The trial sequential monitoring boundary was ignored due to too little information use. The cumulative Z curve did not cross the conventional test boundary (Appendix A, Table 3).

### 3.18. Patient Satisfaction

A total of two studies [7,12] (350 patients) measured patient satisfaction. There was no evidence of a difference between the SA and PNB groups (RR: 1.049; 95% CI 0.981 to 1.122; I^2^ = 25.04; NNTH: 13; 95% CI NNTH −131 to ∞ NNTB 226) (Appendix A, Table 3).

The TSA indicated that 83.7% (350 of 418 patients) of the RIS was accrued. The cumulative Z curve did not cross the conventional test boundary (Appendix A, Table 3).

### 3.19. Quality of Evidence

Eighteen outcomes were evaluated using the GRADE system (Table 4). The quality of the pooled analysis for conversion to general anesthesia was high. The quality of the pooled analysis for block performance time, onset of the sensory block, duration of the motor block, and postoperative analgesics requirement was low. Otherwise, the quality of the pooled analysis was moderate or high.

## 4. Discussion

This meta-analysis of eight RCTs, including 802 patients (of whom 399 patients underwent PNBs and 403 patients underwent SA), demonstrated that the block performance time, onset of sensory and motor blocks, and duration of sensory and motor blocks were significantly shorter, the postoperative analgesics requirements were higher, and the SBP at 30 min was significantly lower in the SA group than in the PNB group. In these outcomes, the cumulative Z curve crossed the trial sequential monitoring boundary, suggesting that the results of TSA reached a sufficient level of evidence and were, therefore, conclusive.

This meta-analysis also showed that PNBs were associated with a decreased incidence of hypotension and vasoactive medications used. However, the cumulative Z curve did not cross the trial sequential monitoring boundary because of sparse data (postoperative analgesic requirements, incidence of hypotension, and vasoactive medications used).

Regarding conversion to general anesthesia, urinary retention, and headache, there was no evidence of differences in the conventional meta-analysis and trial sequential analysis. However, the results from the NNT showed statistically significant differences in these outcomes.

After a brief report that introduced ultrasound-guided regional block techniques in the mid-1990s, the use of this technique has been rapidly increasing [28]. Nerve blocks for foot and/or ankle surgery have been traditionally and commonly used with general anesthesia or SA for postoperative pain relief. PNBs alone neither provide adequate anesthetic coverage for foot and/or ankle surgery nor prevent tourniquet-induced pain. However, two or more combinations of PNBs enable broader anesthetic coverage and prevent tourniquet-induced pain; therefore, foot and/or ankle surgery could be performed under two or more combinations of PNBs. Furthermore, all the studies included in our study compared SA with a combination of two or more PNBs.

In our study, there was no evidence of a difference in the conversion rate to general anesthesia between the PNB and SA groups. Furthermore, our study showed that PNBs provided better pain control, as evidenced by lower postoperative analgesic requirements and longer sensory blocks than SA. PNB also increased the duration of motor block compared with SA. Although there was no evidence of a difference in the meta-analysis and trial sequential analysis, the incidences of PDPH and urinary retention increased regarding the NNT. These findings supported the usefulness of PNBs for foot and/or ankle surgery when two or more types of PNBs were applied. However, the conversion rate to general anesthesia was still high (3.01%) compared with SA (0.66%), which may limit the clinical use of PNBs.

Hypotension is known to result in various side effects, such as cardiac ischemia, cerebral hypoperfusion, acute renal injury, or mortality [29]. In our study, the SBP at 30 min and the incidence of hypotension and use of vasoactive medications were significantly lower in the SA group than in the PNB group. The proposed mechanism of hypotension after SA is reduced systemic vascular resistance (SVR) caused by sympathetic blockage from T1 to L2, decreased cardiac output (CO) caused by the reduced venous return from venous pooling, and decreased cardiovascular compensation mechanisms. PNBs can provide more stable hemodynamics due to a lack of sympathetic nervous system blockage. Therefore, PNBs can be a good choice for patients susceptible to hypotension, such as patients with compromised cardiopulmonary or neurologic function, severe aortic stenosis, or diabetic mellitus [30,31].

However, as shown in our study, the longer performance time and delayed onset of sensory and motor blocks may limit the routine use of PNBs for foot and/or ankle surgery. They may delay the start of surgery and hinder the efficient flow of the operating room. Although this was not observed in our study, technical difficulties, additional needs for equipment such as a peripheral nerve stimulator or ultrasound, and the potential for nerve injury or systemic toxicity due to a relatively larger number of local anesthetics could be barriers to the adoption of PNBs in everyday practice.

This systematic review and meta-analysis with a trial sequential analysis have a number of limitations. First, even after comprehensive and sensitive searching, only nine studies with 802 patients were included in this study. For some outcomes, it may have been underpowered; therefore, the findings from the study were inconclusive. Second, there was clinical and methodological heterogeneity across the studies, which utilized different numbers or types of PNBs and different surgical procedures.

## 5. Conclusions

In our systematic review and meta-analysis with a trial sequential analysis, the block performance time, onset of sensory and motor blocks, and duration of sensory and motor blocks were significantly shorter, postoperative analgesics requirement were higher, and the SBP at 30 min was significantly lower in the SA group than in the PNB group. However, the results for other outcomes are underpowered and, therefore, inconclusive. Thus, to clarify the effectiveness and harm of SA and PNBs, adequately powered and well-planned RCTs are required. Furthermore, when selecting the anesthetic techniques, it is important to take into account the aggressive nature of SA resulting from its central neuroaxial impact, as opposed to PNB.

## Figures and Tables

**Figure 1 jpm-13-01096-f001:**
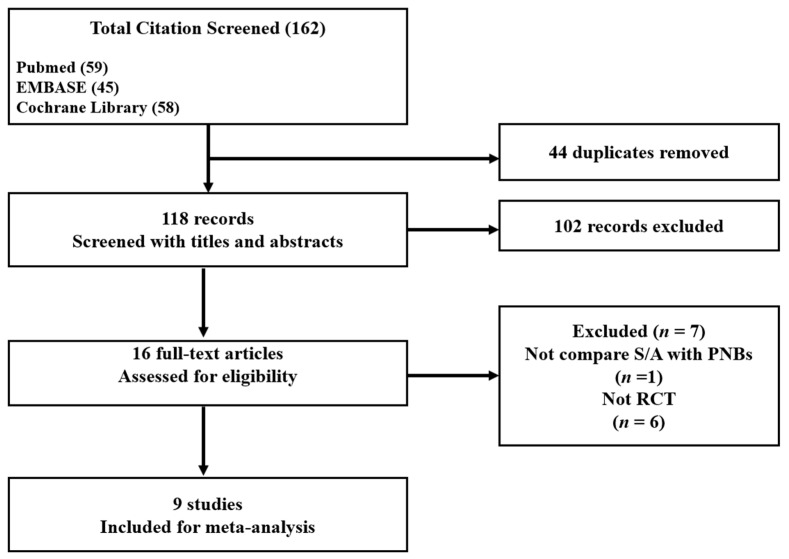
Flow diagram showing the number of abstracts and articles identified and evaluated during the review process.

**Figure 2 jpm-13-01096-f002:**
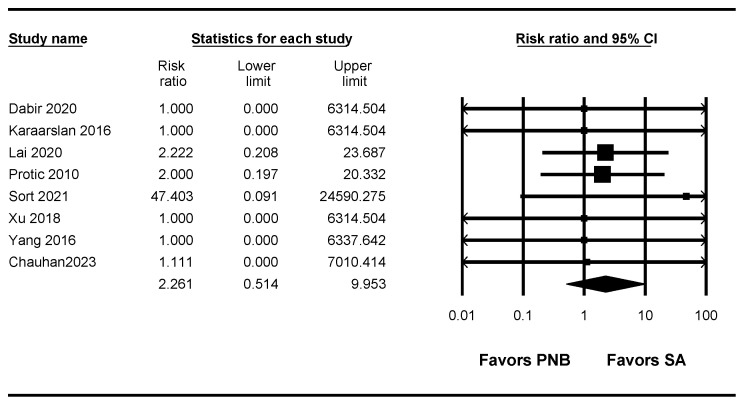
Forest plot showing conversion to general anesthesia. The figure depicts individual trials as filled squares with the relative sample size and the 95% confidence interval (CI) of the difference as a solid line. The diamond shape indicates the pooled estimate and uncertainty for the combined effect [7,8,9,10,11,13,14,15].

**Figure 3 jpm-13-01096-f003:**
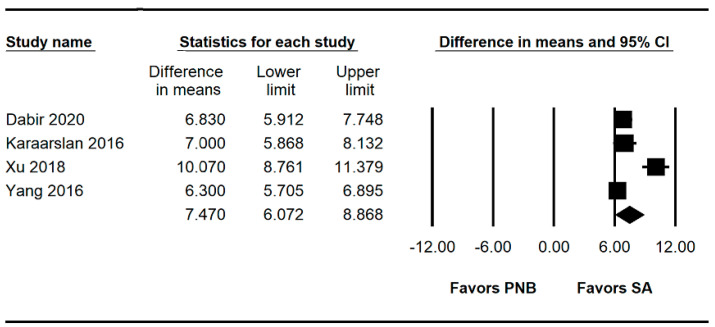
Forest plot showing the block performance time. The figure depicts individual trials as filled squares with relative sample size and the 95% confidence interval (CI) of the difference as a solid line. The diamond shape indicates the pooled estimate and uncertainty for the combined effect. A sensitivity analysis performed by removing one study at a time showed no change in statistical significance (Appendix A). The TSA indicated that the number of accrued patients almost reached the RIS (260 of 262 patients). The cumulative Z curve crossed both the conventional test boundary and the trial sequential monitoring boundary. (Appendix A, Table 3) [8,9,13,14].

**Figure 4 jpm-13-01096-f004:**
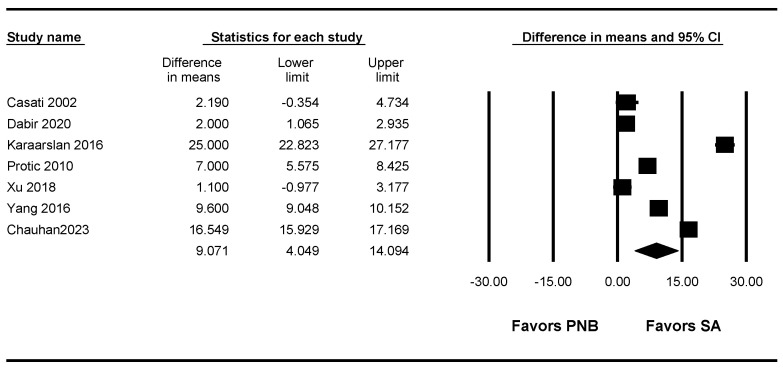
Forest plot showing the onset time of the motor block. The figure depicts individual trials as filled squares with relative sample size and the 95% confidence interval (CI) of the difference as a solid line. The diamond shape indicates the pooled estimate and uncertainty for the combined effect [8,9,11,12,13,14,15].

**Figure 5 jpm-13-01096-f005:**
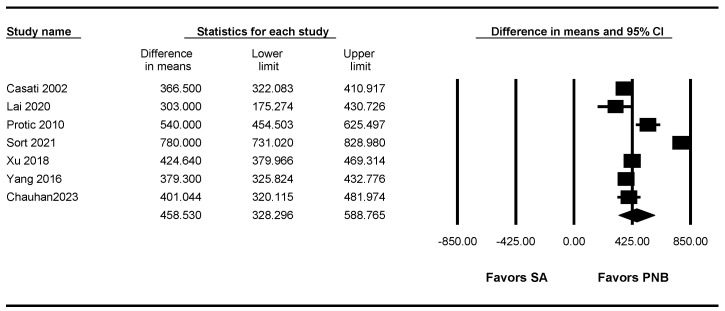
Forest plot showing duration of sensory block. The figure depicts individual trials as filled squares with relative sample size and the 95% confidence interval (CI) of the difference as a solid line. The diamond shape indicates the pooled estimate and uncertainty for the combined effect [7,10,11,12,13,14,15].

**Figure 6 jpm-13-01096-f006:**
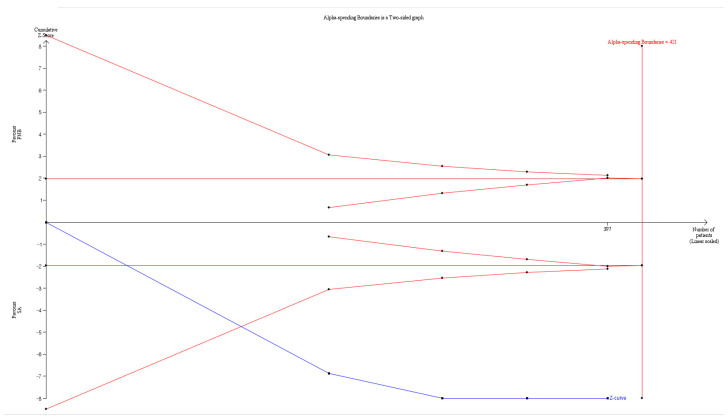
Trial sequential analysis showing the duration of the motor block. The trial sequential analysis (TSA) for the studies comparing the effect of peripheral nerve blocks (PNBs) to that of spinal anesthesia (SA) on the duration of the motor block. The uppermost and lowermost curves represent trial sequential monitoring boundary lines for benefit and harm, respectively. The horizontal line represents the conventional boundaries for statistical significance. The triangular lines on the right side reflect the futility boundaries. The number on the *x*-axis indicates the required information size.

**Table 1 jpm-13-01096-t001:** Characteristics of included studies.

Author, Year, Country	Participants	Sample Size/Intervention	Primary Outcome	Major Findings
Lai et al., 2020, Malaysia [10]	ASA II–III diabetic patients aged >18 years for wound debridement, amputation	PNB (*n* = 45), ultrasound-guided sciatic and femoral or saphenous NB with or without a nerve stimulator SAB (*n* = 50)	Significant hypotension (reduction of ≥30% of SBP)	The SAB group had a large number of patients with significant hypotension.
Chauhan et al., 2023, India [15]	ASA I–III 18–70 years old patients for foot or ankle surgery	PNB (*n* = 27),popliteal sciatic and adductor canal block with nerve stimulator SAB (*n* = 30)	Duration of sensory and motor block	PNB can be alternative technique with advantage of prolonged post-operative analgesia and hemodynamic stability.
Karaarslan et al., 2016, Turkey [9]	ASA I–II 18–60 years old patients for hallux valgus repair	PNB (*n* = 30), ultrasound-guided popliteal sciatic NB with a nerve stimulator and saphenous block as infiltration anesthesia SAB (*n* = 30), unilateral spinal block	Pain VAS at 2 h	VAS at 2, 4, 6, and 12 h were significantly lower, and no adverse effects (hypotension, bradycardia, PDPH, urinary retention) in the PNB group.
Sort et al., 2021, Denmark [7]	ASA I–III patients >18 years old for ankle fracture surgery	PNB (*n* = 77), ultrasound-guided popliteal and saphenous blocks SAB (*n* = 73)	Postoperative pain and opioid consumption	0–27 h of morphine consumption and pain scores were significantly lower in the PNB group.
Dabir et al., 2020, Iran [8]	ASA I–II patients ≥18 for foot or ankle surgery using a pneumatic thigh tourniquet.	PNB (*n* = 30), ultrasound-guided popliteal, femoral, lateral femoral cutaneous blocks SAB (*n* = 30)	Tourniquet pain score	Mean tourniquet pain scores and the total amount of fentanyl and ketamine administered during surgery were significantly higher in the PNB group.
Casati et al., 2002, Italy [12]	ASA I–II patients for foot surgery	PNB (2% mepivacaine, *n* = 50 0.75% ropivacain, *n* = 50), sciatic and femoral NB with nerve stimulator SAB (bilateral, *n* = 50, unilateral, *n* = 50)	Not defined	PNB is an effective and safe as spinal anesthesia with less urinary retention.
Protic et al., 2010, Croatia [11]	Adult trauma patients with bimalleolar fracture	PNB (*n*= 20), ultrasound-guided femoropopliteal block SAB (*n* = 20)	Not defined	PNB provides sufficient anesthesia for ankle fracture.
Xu et al., 2018, China [13]	ASA I–II 20–76 years old patients for hallux valgus surgery	PNB (*n* = 30), ultrasound-guided popliteal and saphenous block SAB (*n* = 30)	Not defined	PNB provides sufficient anesthesia for hallux valgus surgery with maintaining hemodynamic stability.
Yang et al., 2016, China [14]	ASA I–II 18–70 years old patients for ankle surgery	PNB (*n* = 40), ultrasound-guided femoral, obturator, lateral femoral cutaneous and sacral plexus block SAB (*n* = 40)	Not defined	PNB may be safe and effective in patients undergoing ankle surgery.

ASA = American Society of Anesthesiology classification; n = number; PNB = peripheral nerve block; NB = nerve block; SAB = spinal anesthesia block; SBP = systolic blood pressure; VAS = visual analogue scale; PDPH = post-dural puncture headache.

**Table 2 jpm-13-01096-t002:** Risk of bias.

Author, Year	Bias Arising from the Randomization Process	Bias Due to Deviations from Intended Intervention	Bias Due to Missing Outcome Data	Bias in Measurement of the Outcome	Bias in Selection of the Reported Results	Overall Bias
Lai et al., 2020, Malaysia [10]	Low risk	Some concern	Some concern	Low risk	Low risk	High risk
Chauhan et al., 2023, India [15]	Low risk	Some concern	Some concern	Some concern	Low risk	High risk
Karaarslan et al., 2016, Turkey [9]	Some concern	Low risk	Low risk	Low risk	Low risk	Some concern
Sort et al., 2021, Denmark [7]	Low risk	Low risk	Low risk	Low risk	Low risk	Low risk
Dabir et al., 2020, Iran [8]	Some concern	Low risk	Low risk	Low risk	Low risk	Some concern
Casati et al., 2002, Italy [12]	Some concern	Low risk	Low risk	Low risk	Low risk	Some concern
Protic et al., 2010, Croatia [11]	Some concern	Low risk	Low risk	Low risk	Low risk	Some concern
Xu et al., 2018, China [13]	Low risk	Some concern	Low risk	Low risk	Low risk	Some concern
Yang et al., 2016, China [14]	Low risk	Low risk	Low risk	Low risk	Low risk	Low risk

**Table 3 jpm-13-01096-t003:** The summary of the meta-analysis.

	No of Studies	No of Patients	Conventional Meta-Analysis		Trial Sequential Analysis			NNT
			RR or WMD, or SMD with 95% CI	Heterogeneity (I^2^)	Conventional Test Boundary	Monitoring Boundary	RIS	
Conversion to GA	8	602	Not significant (RR: 2.261; 95% CI 0.514 to 9.953)	0.0	Not cross	Not cross	2.9% (602 of 20,771 patients)	Significant (NNTH: 43; 95% CI NNTH 22 to NNTH 478)
Block performance time	4	260	Significant (WMD: 7.470; 95% CI 6.072 to 8.868)	88.66	Cross	Cross	99.2% (260 of 262) patients)	NR
Onset time of sensory block	4	234	Significant (WMD: 7.483; 95% CI 2.837 to 12.130)	99.67	Cross	Cross	18.9% (234 of 1236 patients)	NR
Onset time of motor block	7	554	Significant (WMD: 9.071; 95% CI 4.049 to 14.094)	99.40	Cross	Cross	93.7% (554 of 591 patients)	NR
Duration of sensory block	7	679	Significant (WMD: 458.53; 95% CI 328.296 to 588.765)	97.02	Cross	Cross	exceeds RIS (679 of 246 patients)	NR
Duration of motor block	4	397	Significant (WMD: 247.416; 95% CI 95.625 to 399.208)	97.325	Cross	Cross	94.3% (397 of 421 patients)	NR
Postoperative analgesics requirement	3	407	Significant (SMD: −1.091; 95% CI −1.634 to −0.549)	83.10	Cross	Cross	23.4% (407 of 1741 patients)	NR
Incidence of hypotension	4	272	Significant (RR: 0.152; 95% CI 0.042 to 0.548)	0.0	Cross	Not cross	6.7% (272 of 4043 patients)	Significant (NNTB: 7; 95% CI NNTB 5 to NNTH 14)
Vasoactive drug	3	305	Significant (RR: 0.253; 95% CI 0.101 to 0.638)	0.0	Cross	Not cross	8.2% (305 of 3698 patients)	Significant (NNTB: 8; 95% CI NNTB 5 to NNTH 15)
SBP at baseline (T0)	3	212	Not significant (WMD: −0.112; 95% CI −5.329 to 5.168)	31.501	Cross	Not cross	1.6% (212 of 9127 patients)	NR
SBP at 30 min after the beginning of surgery (T30)	3	212	Significant (WMD: 13.950; 95% CI 4.603 to 23.298)	74.829	Cross	Cross	exceeds RIS (212 of 75 patients)	NR
DBP at baseline (T0)	3	212	Not significant (WMD: −1.297; 95% CI −3.931 to 1.337)	0.0	Not cross	Not cross	11.8% (212 of 1802 patients)	NR
DBP at 30 min after the beginning of surgery (T30)	3	212	Not significant (WMD: 4.535; 95% CI −0.969 to 10.039)	75.674	Not cross	Not cross	32.8 (212 of 646 patients)	NR
Heart rate (T0)	3	212	Not significant (WMD: −0.204; 95% CI −3.696 to 3.288)	0.0621	Not cross	Not cross	11.8% (212 of 1802 patients)	NR
Heart rate (T30)	3	212	Not significant (WMD: 2.617; 95% CI −3.265 to 8.599)	66.401	Cross	Not cross	9.5% (212 of 2237 patients)	NR
Urinary retention	4	354	Not significant (RR: 0.089; 95% CI 0.002 to 3.297)	0.0	Not cross	Not cross	18.5% (354 of 1912 patients	Significant NNTB: 18; 95% CI NNTB 11 to NNTH 45
PDPH	4	234	Not significant (RR: 0.159; 95% CI 0.004 to 6.071)	0.0	Not cross	Not cross	1.2% (234 of 14,721 patients)	Significant (NNTB: 30; 95% CI NNTB 15 to NNTH 813)
Patients’ satisfaction	2	350	Not significant (RR: 1.047; 95% CI 0.989 to 1.108)	25.04	Not cross	Not cross	83.7% (350 of 418 patients)	Not significant (NNTH: 13; 95% CI NNTH −131 to ∞ to NNTB 226)

No; number, RR; relative risk, WMD; weighted mean difference, SMD; standardized mean difference, CI; confidence interval, NNT; number needed to treat; RIS; required information size, GA; general anesthesia, NNTH; number needed to treat harm, NNTB; number needed to treat benefit, NR; not reported, SBP; systolic blood pressure, DBP; diastolic blood pressure, PDPH; post-dural puncture headache.

**Table 4 jpm-13-01096-t004:** The GRADE evidence quality for each outcome.

Outcomes	Number of Studies	Quality Assessment					Quality
		ROB	Inconsistency	Indirectness	Imprecision	Publication Bias	
Conversion to GA	8	not serious	not serious	not serious	not serious	NA	⨁⨁⨁⨁ High
Block performance time	4	serious	serious	not serious	not serious	NA	⨁⨁◯◯ Low
The onset of sensory block	4	not serious	serious	not serious	serious	NA	⨁⨁◯◯ Low
The onset of motor block	7	not serious	serious	not serious	not serious	NA	⨁⨁⨁◯ Moderate
Duration of sensory block	7	not serious	serious	not serious	not serious	NA	⨁⨁⨁◯ Moderate
Duration of motor block	4	not serious	serious	not serious	serious	NA	⨁⨁◯◯ Low
Postoperative analgesics requirement	3	not serious	serious	not serious	serious	NA	⨁⨁◯◯ Low
Incidence of hypotension	4	serious	not serious	not serious	not serious	NA	⨁⨁⨁◯ Moderate
Vasoactive drug	3	not serious	not serious	not serious	serious	NA	⨁⨁⨁◯ Moderate
SBP at baseline (T0)	3	not serious	not serious	not serious	serious	NA	⨁⨁⨁◯ Moderate
SBP at 30 min after the beginning of surgery (T30)	3	not serious	not serious	not serious	serious	NA	⨁⨁⨁◯ Moderate
DBP at baseline (T0)	3	not serious	not serious	not serious	serious	NA	⨁⨁⨁◯ Moderate
DBP at 30 min after the beginning of surgery (T30)	3	not serious	not serious	not serious	serious	NA	⨁⨁⨁◯ Moderate
Heart rate (T0)	3	not serious	not serious	not serious	serious	NA	⨁⨁⨁◯ Moderate
Heart rate (T30)	3	not serious	not serious	not serious	serious	NA	⨁⨁⨁◯ Moderate
Urinary retention	4	not serious	not serious	not serious	serious	NA	⨁⨁⨁◯ Moderate
PDPH	4	not serious	not serious	not serious	serious	NA	⨁⨁⨁◯ Moderate
Patient satisfaction	2	not serious	not serious	not serious	serious	NA	⨁⨁⨁◯ Moderate

ROB; Risk of bias. GA; general anesthesia, NA; not applicable. SBP; systolic blood pressure, DBP; diastolic blood pressure, PDPH; post-dural puncture headache. Quality are rated with 4 grades: very low (⨁◯◯◯), low (⨁⨁◯◯), moderate (⨁⨁⨁◯), high (⨁⨁⨁⨁).

## Data Availability

Data are contained within the article and Appendix A.

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
