# Peer review of "Comparison of a Peripheral Nerve Block versus Spinal Anesthesia in Foot or Ankle Surgery: A Systematic Review and Meta-Analysis with a Trial Sequential Analysis"

_jpm, 2023, doi:10.3390/jpm13071096_

Round 1
Reviewer 1 Report
Dear authors please provide a more precise conclusion for this extensive work. It is difficult to compare these two techniques because of the nature of the blocks, the SA is more agressive in lot of aspects and due to the "central" neuroaxial effect is not similar to peripheral blocks. Best regards
Dear Editor
My appologies for a delayed review.
Author Response
Thank you for reviewer's comment for the betterment of manuscript. According to reviewer's comment, we revised the manuscript.

Reviewer 2 Report
Dear authors,
Thank you for submitting your work. I congratulate all the authors for compiling and submitting this review article.
The article is well written and follows the instructions of a systematic review and meta-analysis.
However, I do not have access to the supplementary files mentioned by you (especially the TSA graphs). During revision, I would like to see the TSA graphs and see the correlation.
The overall English language is fine and will just need technical edits (subject to acceptance).
Author Response
Thank you for reviewer's support and comments.
According to reviewer's comment, I attached the supplementary material here
